# Coping Strategies and Health-Related Quality of Life in Pregnant Women with SARS-CoV-2 Infection

**DOI:** 10.3390/medicina57101113

**Published:** 2021-10-16

**Authors:** Madalina Timircan, Felix Bratosin, Iulia Vidican, Oana Suciu, Mirela Turaiche, Adrian Vasile Bota, Sorina Mitrescu, Iosif Marincu

**Affiliations:** 1Department of Gynecology, “Victor Babes” University of Medicine and Pharmacy, 300041 Timisoara, Romania; timircan.madalina@yahoo.com; 2Methodological and Infectious Diseases Research Center, Department of Infectious Diseases, “Victor Babes” University of Medicine and Pharmacy, 300041 Timisoara, Romania; iulia.georgianabogdan@gmail.com (I.V.); suciu.oana@umft.ro (O.S.); mirela.turaiche@gmail.com (M.T.); bota.adrian1@yahoo.com (A.V.B.); sorina.mitrescu@yahoo.ro (S.M.); imarincu@umft.ro (I.M.)

**Keywords:** SARS-CoV-2, COVID-19, pregnancy, HADS, CORE-OM, SF-12, COPE, QPP

## Abstract

*Background and Objectives:* As maternal deaths associated with the SARS-CoV-2 infection remain at several times greater than the general population, significant factors that might contribute to the higher mortality and morbidity rate are the psychological impact of the disease and pregnancy itself. Therefore, the current study’s main objective was to assess how pregnant women react and cope with the stress of COVID-19 disease and how it influences their overall health and quality of life in healthcare facilities. *Materials and Methods:* In this cross-sectional study, we included 304 pregnant women who successfully completed standardized forms to assess our topics of interest, comprising of the Hospital Anxiety and Depression Scale, the Short Form Health Survey-12, the Coping Orientation to Problems Experienced Inventory scale, the CORE-Outcome Measure Questionnaire, and the Quality from the Patient’s Perspective questionnaire. *Results*: Unemployed, pregnant women living in poverty in the rural areas had higher SARS-CoV-2 infection rates during pregnancy. They faced higher anxiety levels and depression rates, with associated increased physical burden and exhaustion. However, these findings are not influenced by hospital care since it remained unchanged among COVID-19 and non-COVID-19 maternity units, excepting significantly lower technical competence scores of COVID-19 facilities. *Conclusions*: As the pandemic’s consequences emerge and additional outbreaks occur, care must prioritize the additional physical burden experienced by pregnant women who have contracted COVID-19, as well as psychological, emotional, and mental health support.

## 1. Introduction

In December 2019, a previously undiscovered cause of pneumonia was found in the city Wuhan from the Hubei region in China, dubbed as the severe acute respiratory syndrome coronavirus 2 (SARS-CoV-2), causing the coronavirus disease (COVID-19) [1]. The COVID-19 illness rapidly escalated into a major public health catastrophe, eventually reaching pandemic proportions [2]. COVID-19’s effect has been felt most acutely in vulnerable and high-risk populations, such as pregnant women, due to its increasing global prevalence and persistent spikes [3]. Immunocompromised women, pregnant women, and members of African-American and Asian descent and ethnic minority groups were proven to be more susceptible to COVID-19’s medical and psychosocial consequences [4].

Recent data suggest that, although very uncommon, in utero vertical transmission of SARS-CoV-2 is feasible [5]. COVID-19 infection is as severe in pregnant women as in non-pregnant people, in particular cases increasing the risk of spontaneous abortion or premature labor [6]. Although pregnant women are less impacted by COVID-19 than the general population, they may be more susceptible to a severe form of the disease if they are in respiratory failure. Previous research has shown the broad and severe effect of epidemics on people’s mental health, which may result in the development of new psychiatric symptoms or the aggravation of pre-existing mental disease [7]. COVID-19, as a public health emergency, has generated widespread worry and psychological distress among pregnant women and their spouses. Perinatal anxiety is a new complication of the pandemic, with serious consequences for physical health. COVID-19’s effect on pregnant women is direct and indirect, as a result of lockdown, social distance, and isolation measures used as part of worldwide attempts to contain the virus’s spread.

Previous research on the SARS epidemic has demonstrated that pregnant women are more prone to anxiety than non-pregnant women [8]; this includes anxiety about infection, the transmission of infection to the fetus, acquired infection during childbirth, and teratogenicity of microorganisms and medications. They avoided hospitals and health care facilities out of SARS-CoV-2 infection fear and caused delayed prenatal treatment. Pregnant women who have COVID-19 are recommended to seek specialized treatment for diagnosis, management, and avoidance of problems for both mother and baby [9]. It is very probable that birth problems such as low birth weight, shorter gestational age, vomiting throughout pregnancy, preeclampsia, poor Apgar scores, and prolonged hospital stay have been linked to perinatal anxiety and depression, as well as affecting the overall quality of life [10]. The terms “quality of life” and “quality of care” are often used to describe both the health-related quality of life and the woman’s perception of the quality of care received in healthcare settings. Due to the high prevalence of COVID-19 and its negative psychological effects on people’s lives, as well as an absence of research on the psychological well-being of pregnant women during the COVID-19 pandemic, it is necessary to assess the impact of COVID-19 on the mental health status of pregnant and lactating women.

The purpose of this research was to assess and evaluate psychosocial changes and coping strategies associated with COVID-19 infection and its effect on health-related quality of life in pregnant women affected by SARS-CoV-2 during pregnancy. We planned to gain insight into pregnant women’s physical and mental health and to assess the quality of medical care provided during the COVID-19 pandemic. Five standardized questionnaires were used: the Hospital Anxiety and Depression Scale (HADS), the Short-Form Health Survey (SF-12), the Coping Orientation to Problems Experienced Inventory scale (COPE), the CORE-Outcome Measure Questionnaire (CORE-OM), and the Quality from the Patient’s Perspective questionnaire (QPP).

## 2. Materials and Methods

### 2.1. Study Design and Participants

The present study took place in an outpatient setting, in a joint collaboration between the University Clinic of Obstetrics and Gynecology “Bega” and “Dr. Victor Babes” Infectious Diseases and Pulmonology Clinical Hospital from Timisoara, Romania. Our research followed a cross-sectional design, and we opted for a convenience sampling technique to calculate the optimal sample size, which was estimated to include at least 377 individuals for a confidence coefficient of 95%. A total of 108 women were removed from the research out of 412 who accepted to participate in the study and complete our questionnaires, owing to a lack of entrance requirements or insufficient completion of surveys, leaving for a total of 304 participants. This research included pregnant women who suffered from the SARS-CoV-2 infection (168 cases) and women who carried pregnancies during the pandemic but without being confirmed with COVID-19 (136 cases). The questionnaires were filled between April and May 2021. At the time of study the COVID-19 vaccination campaign was in its first three months, where healthcare workers, older people, and other special categories of patients with comorbid conditions were the first to benefit. Therefore, vaccination among pregnant women was not recommended in Romania, since there were no special directives from the Ministry of Health, while obstetricians and pregnant women were reluctant to receive the new vaccines. As consequence, there was no vaccinated patient included in our study.

To ensure physical distance was maintained in order to avoid the spread of COVID-19, participants were able to view an online version of the questionnaire that was identical in terms of questions, words, and presentation sequence. Members of the study team sent the questionnaire links to pregnant women through email, while the online database’s information sorting procedure was entirely automated, and each participant may answer questions electronically only once. The survey was used to gather data on participants’ demographics, obstetrical features, depression, anxiety, quality of life, and coping strategies during stressful situations. Study participants were asked to fill the following surveys: (1) the Hospital Anxiety and Depression Scale (HADS); (2) the Short Form Health Survey (SF-12); (3) the Coping Orientation to Problems Experienced Inventory scale (COPE); (4) the CORE-Outcome Measure Questionnaire (CORE-OM); and (5) the Quality from the Patient’s Perspective questionnaire (QPP).

### 2.2. Variables

The COPE Inventory is a 60-questions survey that evaluates 15 different coping mechanisms for stress [11]. Each item may be rated on a scale of 1 to 4, with one indicating “I seldom do this” and four indicating “I often do this.” Except for mental disengagement, behavioral disengagement, denial, and drug use, the other coping techniques cannot be classified as “good” or “bad,” but rather as distinct methods of dealing with stressful circumstances. There are many classifications for coping techniques. One of them relates to how an individual responds to stress: engagement vs. disengagement. The engagement coping cluster encompasses the strategies an individual employs to cope with a stressor or associated emotions, including positive reinterpretation and growth, emotion focus and venting, instrumental social support, active coping, religious coping, humor, emotional and social support, acceptance, suppression of competing activities, and planning. By contrast, the coping disengagement cluster refers to the methods that an individual does to avoid experiencing stress and the associated emotions such as mental disengagement, behavioral disengagement, denial, and substance use. When engaging or disengaging stress, one may handle the stressor directly (problem-focused coping) or indirectly via its associated emotions—the discomfort (emotion-focused coping). The emotion-focused cluster includes the following strategies: emotional attention and venting, instrumental social support, and drug usage. The cluster of problem-focused coping methods includes positive reinterpretation and development, religious coping, humor, suppression of competing activities, and planning.

The HADS test is a 14-item instrument [12] used to test for depression and anxiety, where seven questions are meant for measuring depression and the other seven for measuring anxiety. Increased scores indicate increased anxiety and depression symptoms, while a score of 11 or more is considered a clinical illness.

The SF-12 is a commonly used instrument for evaluating overall health and health outcomes [13]; it was formerly used to evaluate health-related quality of life. The SF-12 is a 12-item questionnaire that assesses physical and mental health. The physical and mental component summary scores were calculated in accordance with established standards. A low score indicates poor physical or mental health.

The CORE-OM is a 34-item validated self-report questionnaire that utilizes a five-point scale ranging from “not at all” to “most of/all of the time” [14]. Women’s well-being, issues and symptoms, daily functioning, and risk/harm are all recorded in four dimensions. To measure the degree of global psychological distress, mean and total scores were computed. A higher score indicates better health and less global distress in terms of well-being, issues and symptoms, and daily functioning. A high risk or harm score indicates a greater degree of psychological discomfort.

The QPP is a 24-question model for assessing a woman’s perception and experience of the quality of care in a healthcare setting [15]. It is composed of four dimensions: the healthcare provider’s medical-technical competence, the healthcare organization’s physical-technical conditions, the caregivers’ degree of identity orientation in their attitudes and actions, and the socio-cultural atmosphere of the healthcare setting. A Likert scale (1–4) was used to answer questions ranging from “do not agree at all” to “somewhat agree”, “mainly agree”, and “totally agree”.

We compared the questionnaire results for stress-related coping strategies and health-related quality of life of SARS-CoV-2-infected pregnant women with a control group of pregnant women without COVID-19, using the IBM SPSS v.26 statistical software. The χ^2^ test and Fisher exact test were used for categorical variables and Student’s *t*-test or Mann-Whitney U-test for continuous variables. The significance threshold was set for α = 0.05.

### 2.3. Ethics

The Local Committee of Ethics for Scientific Research of “Dr. Victor Babes” Infectious Diseases and Pulmonology Clinical Hospital Timisoara operates under art. provisions 167 of Law no. 95/2006, art. 28, chapter VIII of order 904/2006 and with EU GCP Directives 2005/28/EC, International Conference on Harmonization of Technical Requirements for Registration of Pharmaceuticals for Human Use (ICH), and with the Declaration of Helsinki—Recommendations Guiding Medical Doctors in Biomedical Research Involving Human Subjects. The current study protocol received ethical approval on the 28 February 2021, with approval number 1626. Women who visited a maternity unit during the COVID-19 epidemic and were at least 18 years old were eligible to enroll. All patients included in the study agreed to be involved by signing a standardized informed consent form available in our clinic.

## 3. Results

The study collected data from 304 patients, as COVID-19 and non-COVID-19 groups, with 168 pregnant women who suffered from the SARS-CoV-2 infection in the COVID-19 group, and 136 women who were pregnant during the pandemic but not infected with SARS-CoV-2 and had at least one hospital admission for pregnancy follow-up or other medical reasons, in the non-COVID-19 group. Our patients were grouped by level of income into low, medium, and high, with a statistically significant difference between proportions (*p*-value = 0.048). The biggest difference was observed in the population of low-income patients infected with the SARS-CoV-2 group, as compared to the non-infected patients (22.6% vs. 12.5%). Another grouping variable was the occupation status, patients being evaluated as employed, self-employed, and unemployed, with a statistically significant difference in proportions between COVID-19 and non-COVID-19 groups (*p*-value = 0.039). A total of 20.9% of our pregnant women with SARS-CoV-2 infection were unemployed, as compared with a level of unemployment of just 11.8% in the other group. Lastly, we observed that 61.9% of patients in the COVID-19 group had an urban place of origin versus 72.8% of patients in the non-COVID-19 group (*p*-value = 0.045). Age, civil status, parity, and level of education did not significantly differ between the study groups (Table 1).

Based on the COPE-60 questionnaire (Table 2, Figure 1), we observed a significant difference in coping strategies based on the presence or absence of the SARS-CoV-2 infection. Apparently, infected pregnant women tend towards a high likelihood of using a disengagement method of coping with stress in a 54.7% proportion, versus 43.3% in the other women who carried a pregnancy during the pandemic but didn’t get infected (*p*-value = 0.048). Other significant findings were identified at the level of positive coping methods, having 48.6% of patients in the non-COVID-19 group apply an engagement coping technique, compared to the COVID-19 pregnancies group, where only 33.3% of women used this positive coping method (*p*-value = 0.007). Patients in the COVID-19 positive group used emotion-focused coping mechanisms at a 38.1% high likelihood, compared with the non-COVID-19 group with a 52.2% high likelihood (*p*-value = 0.013).

Based on the Hospital Anxiety and Depression Scale (Table 3, Figure 2), there was a statistically significant difference between our study groups concerning the depression, anxiety, and total HADS scores. Pregnant women without SARS-CoV-2 infection throughout their pregnancy scored significantly lower scores on all three measures (*p*-value < 0.001).

The SF-12 questionnaire (Table 4, Figure 3) described significantly higher differences in physical health and general health functionality (total score) in the non-COVID-19 pregnant women (59.7 vs. 44.1, *p*-value < 0.001), respectively 55.5 vs. 47.7, *p*-value = 0.004. On the sub-group level, the COVID-19 pregnant women were more likely to be limited by pain interfering with their normal functioning (2.04 vs. 1.26, *p*-value = 0.038) and more likely to feel sad (2.11 vs. 1.33, *p*-value = 0.040).

We did not observe any significant differences between the two study groups based on the CORE-OM results (Table 5, Figure 4), although patients in the COVID-19 group were more likely to lack energy and enthusiasm (1.3 vs. 2.2, *p*-value = 0.041), and complaining of more frequent physical problems (1.1 vs. 2.1, *p*-value = 0.038), as compared to the other patients in the non-COVID-19 group.

Based on the QPP questionnaire results (Table 6, Figure 5), the pregnant women infected with SARS-CoV-2 reported overall lower scores in regards to their perceived hospital care and conditions, with the only significant difference compared to the non-COVID-19 group being observed in the medical, technical competence score (3.16 vs. 3.94, *p*-value = 0.036).

## 4. Discussion

The World Health Organization (WHO) has recommended isolation to prevent the human spread of the SARS-CoV-2 because of its high infectiousness and rise in incidence [16]. Numerous signs of mental discomforts, such as sadness, tension, irritability, and sleeplessness, have been more prevalent among individuals confined in isolation. Quarantine stressors include extended quarantine duration, fear of infection, disruption of normal life routines, decreased social activity and physical contact with others, insufficient basic supplies, a lack of sufficient information and clear instructions on how to proceed, and severe socioeconomic problems [17].

Previous SARS pandemic studies have shown that pregnant women, particularly those who are more emotionally sensitive, have elevated levels of anxiety [18]. Stress and anxiety impair the immune system and predispose individuals to infectious illnesses, and many studies have shown a link between mental morbidity during pregnancy and poor pregnancy outcomes such as low birth weight and premature labor. Additionally, both prenatal and postpartum depression are linked with decreased breastfeeding length [19], and the mother’s anxiety is connected with breastfeeding problems, decreased breastfeeding intention, and decreased breastfeeding duration [20]. The cumulative impact of the mental load placed on society by COVID-19, along with the fact that pregnancy and lactation are psychologically sensitive periods, may account for these findings. Given the debilitating effects of anxiety and depression on the immune system, pregnancy, and lactation, our findings highlight the critical role of mental health treatment for pregnant and breastfeeding women in extending the length of outbreaks. Our findings indicate that women who had COVID-19 during the pandemic era had a substantially greater physical burden and consequential mental burden, causing high levels of anxiety and depression. It is unknown how the exact pathophysiology of COVID-19 in pregnancy may be both milder in general among pregnant women and very severe in individuals with acute severe respiratory failure. Several theories include the protective hormonal environment lessening the severity of the illness, comparable to the hormonal effects seen during influenza infections during pregnancy. Another possibility is that the immune response to viral infections promotes the production of anti-inflammatory cytokines, which may mitigate the severity of COVID-19 in pregnancy [21]. While much remains unclear in spite of these theories, our COVID-19 sample reported an acute physical load on their functional daily physical ability.

Global studies have noted a rise in anxiety relative to pre-pandemic levels, including an increase in sadness and anxiety among pregnant women and their spouses [22]. We observed no significant difference in health-related quality of life between the COVID-19 and non-COVID-19 groups of women in terms of their mental health and psychological well-being. Both groups reported comparable psychological well-being ratings on two validated measures, the SF-12 and the CORE-OM, and those scores were highly associated, indicating the validity of the assessments. However, the COVID-19 affected group is more likely to lack energy throughout the day and complain of more frequent physical problems than the non-COVID-19 pregnant women. Comparable research on the psychological impact of COVID-19 among pregnant women conducted in the United Kingdom [23] found no significant difference in anxiety levels between pregnant women who had or did not have COVID-19. However, this research had access to a limited sample size and used a variety of verified techniques that were different than ours, such as the Generalized Anxiety Score 7 (GAD-7) and Patient Health Questionnaire-9 (PHQ-9). It discovered that sociodemographic factors may have had a larger effect on perinatal women’s mental health during the pandemic. Our results corroborate previous findings, indicating that the anxiety and mental health effects seen during the COVID-19 pandemic result from the cultural and social context, not the SARS-CoV-2 infection itself. It is likely that possible differences in results obtained between the UK study and ours are explained by the difference in demographics, considering the Romanian population is ethnically homogenous and a significantly larger population of Romanians live in rural areas, where the most significant differences were observed in our study.

Considering our findings and on the basis of other studies conducted on the topic of brain metabolism and biochemical alterations in stress [24,25], a future direction for research based on the topic discussed here could be an evaluation and correlation of biochemical reactions and neurotransmitters involved in stress adaptation and the severity of trauma in pregnant patients with COVID-19 infection. COVID-19 positive pregnant women in our study scored substantially worse on both depression and anxiety aspects of mental health than the pregnant women who did not experience the burden of being infected with SARS-CoV-2 during their pregnancy and requirement of hospitalization. They also tend to use maladaptive coping mechanisms. These findings seem to corroborate previous studies [26] indicating an effect of COVID-19 knowledge on the rising incidence of prenatal depression, as its incidence rose as the number of deaths and newly diagnosed patients increased, as did the lack of knowledge about the disease’s characteristics, fear of infection, and vertical transmission from mother to fetus.

## 5. Conclusions

COVID-19 treatment in pregnant women continues to be a problem for obstetricians and doctors. During these periods, it is critical that pregnant women receive holistic care. As the pandemic’s consequences emerge and additional outbreaks occur, care must prioritize the additional physical burden experienced by pregnant women who have contracted COVID-19, as well as psychological, emotional, and mental health support. The maternities must continue to fulfill the necessary standards for anticipated quality of care while caring for mothers, babies, and their families throughout this continuing crisis. There is a genuine risk of increased maternal, fetal, and pregnancy problems in unsupported laboring and pregnant women, with the research emphasizing the critical nature of social relationships throughout pregnancy.

Specific findings by our study identified several pregnancy risk groups that are more impacted by the stress of COVID-19, who are prone to use negative coping strategies, degrading their psychic recovery and quality of life. Especially unemployed pregnant women living in poverty in the rural areas seem at greater risk of SARS-CoV-2 infection during pregnancy, with the associated mental challenges such as higher anxiety and depression rates or increased physical burden. Finally, the quality of hospital care remains unchanged among COVID-19 and non-COVID-19 maternity units, excepting the lower technical competence of COVID-19 facilities, which can be attributed to the lack of experience in such scenarios and constant change of regulations to patient care.

## Figures and Tables

**Figure 1 medicina-57-01113-f001:**
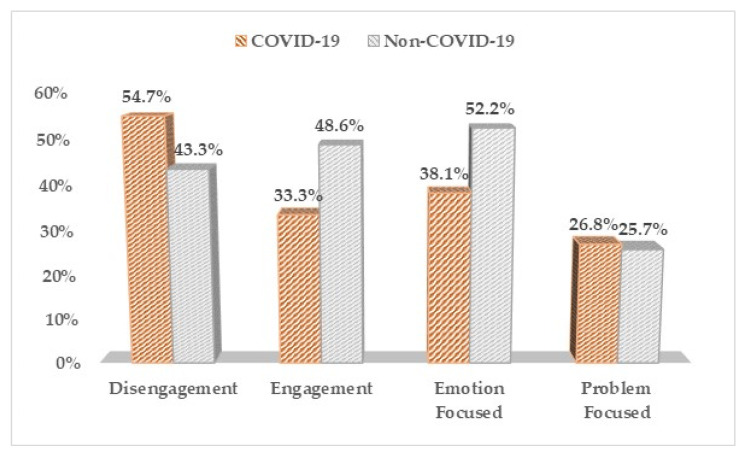
Comparison between COVID-19 positive and COVID-19 negative pregnant women based on their high likelihood of using the coping mechanisms determined by the COPE-60 survey.

**Figure 2 medicina-57-01113-f002:**
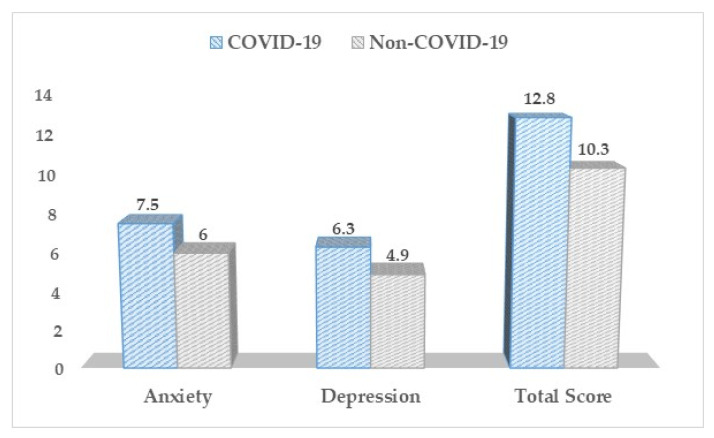
HADS questionnaire results between COVID-19 positive and negative pregnant women.

**Figure 3 medicina-57-01113-f003:**
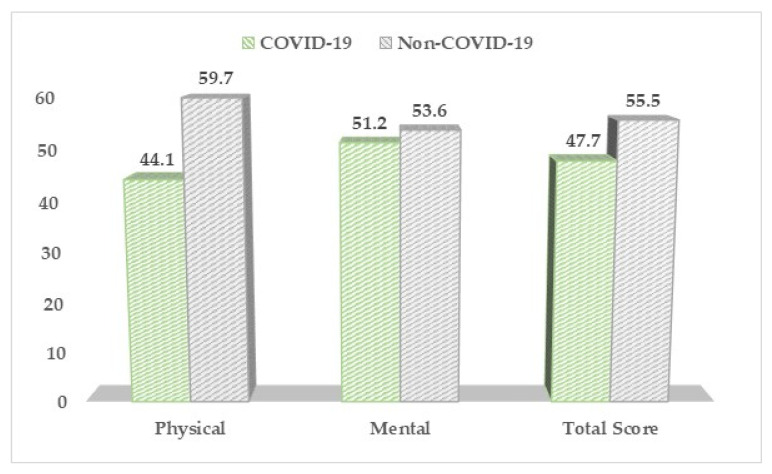
SF-12 questionnaire results between COVID-19 positive and negative pregnant women.

**Figure 4 medicina-57-01113-f004:**
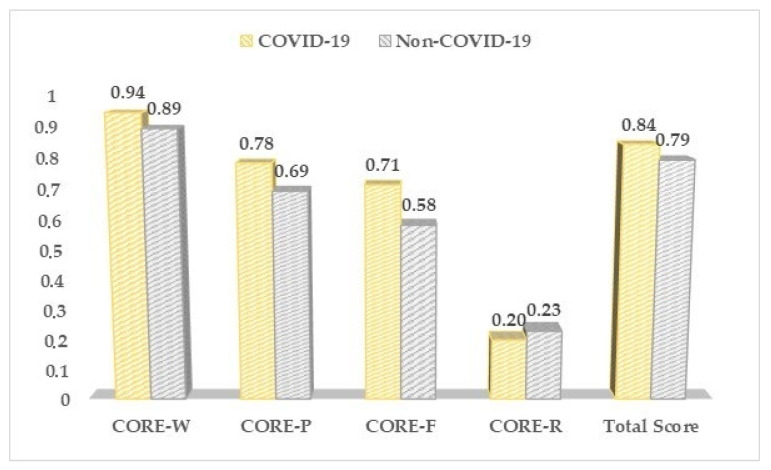
CORE-OM questionnaire results between COVID-19 positive and negative pregnant women.

**Figure 5 medicina-57-01113-f005:**
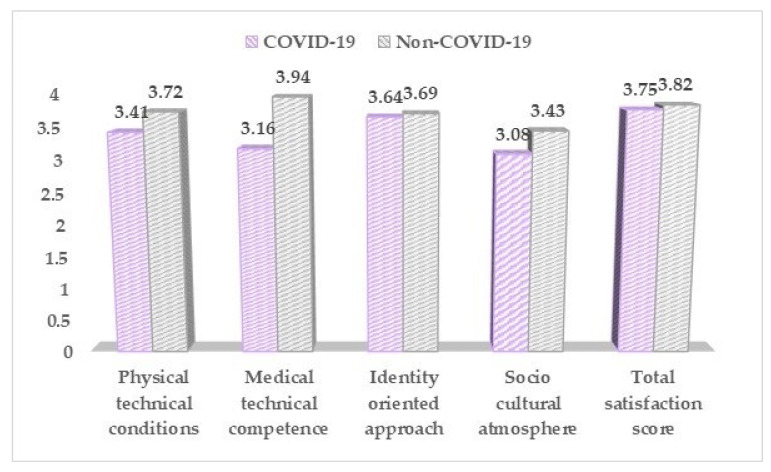
QPP questionnaire results between COVID-19 positive and negative pregnant women.

**Table 1 medicina-57-01113-t001:** General characteristics of the study groups.

Variables (Frequencies) *	COVID-19 (*n* = 168)	Non-COVID-19 (*n* = 136)	*p*-Value
**Age** (mean ± SD)	31.4 ± 7.8	30.9 ± 6.4	0.693
**Civil Status**			0.877
Married	153(91.0%)	127(93.3%)	
Single	4(2.3%)	3(2.2%)	
Divorced	7(4.1%)	4(2.9%)	
Widowed	4(2.3%)	2(1.4%)	
**Parity**			0.220
Primigravida	39(23.2%)	40(29.4%)	
Multiparous	129(76.8%)	96(70.5%)	
**Income**			0.048
Low	38(22.6%)	17(12.5%)	
Medium	86(51.2%)	72(52.9%)	
High	44(26.2%)	47(34.5%)	
**Education**			0.548
≤12 years	42(25.0%)	30(22.1%)	
>12 years	126(75.0%)	106(77.9%)	
**Occupation**			0.039
Employed	88(52.3%)	69(50.7%)	
Self-Employed	45(26.8%)	51(37.5%)	
Unemployed	35(20.9%)	16(11.8%)	
**Place of Origin**			0.045
Rural	64(38.1%)	37(27.2%)	
Urban	104(61.9%)	99(72.8%)	

* Unless described differently.

**Table 2 medicina-57-01113-t002:** COPE-60 questionnaire results between COVID-19 positive and negative pregnant women.

COPE-60	Likelihood *	COVID-19 (*n* = 168)	Non-COVID-19 (*n* = 136)	*p*-Value
Disengagement				0.048
	High (3–4)	92(54.7%)	59(43.3%)	
	Low (1–2)	76(45.3%)	77(56.6%)	
Engagement				0.007
	High (3–4)	56(33.3%)	66(48.6%)	
	Low (1–2)	112(66.7%)	70(51.4%)	
Emotion Focused				0.013
	High (3–4)	64(38.1%)	71(52.2%)	
	Low (1–2)	104(61.9%)	65(47.8%)	
Problem Focused				0.836
	High (3–4)	45(26.8%)	35(25.7%)	
	Low (1–2)	123(73.2%)	101(74.3%)	

* High likelihood of using the coping mechanism: answers 3 and 4; Low likelihood of using the coping mechanism: answers 1 and 2.

**Table 3 medicina-57-01113-t003:** HADS questionnaire results between COVID-19 positive and negative pregnant women.

HADS (Mean ± SD)	COVID-19 (*n* = 168)	Non-COVID-19 (*n* = 136)	*p*-Value
Anxiety	7.5 ± 4.3	6.0 ± 4.4	<0.001
Depression	6.3 ± 3.5	4.9 ± 3.1	<0.001
Total Score	12.8 ± 6.1	10.3 ± 5.5	<0.001

**Table 4 medicina-57-01113-t004:** The quality of life physical and mental scores based on the SF-12 survey results between COVID-19 positive and negative pregnant women.

Physical and Mental Health (Mean ± SD)	COVID-19 (*n* = 168)	Non-COVID-19 (*n* = 136)	*p*-Value
Physical	44.1 ± 9.5	59.7 ± 8.1	<0.001
Mental	51.2 ± 9.3	53.6 ± 10.4	0.196
Total Score	47.7 ± 8.2	55.5 ± 7.6	0.004

**Table 5 medicina-57-01113-t005:** The CORE-OM results between COVID-19 positive and negative pregnant women.

CORE-OM (Mean ± SD) *	COVID-19 (*n* = 168)	Non-COVID-19 (*n* = 136)	*p*-Value
CORE-W	0.94 ± 0.72	0.89 ± 0.69	0.652
CORE-P	0.78 ± 0.60	0.69 ± 0.57	0.188
CORE-F	0.71 ± 0.55	0.58 ± 0.44	0.054
CORE-R	0.20 ± 0.13	0.23 ± 0.14	0.747
Total Score	0.84 ± 0.61	0.79 ± 0.58	0.841

* Clinical Outcomes—CORE W: well-being; CORE P: problems and symptoms; CORE F: life functioning; CORE R: risk and harm.

**Table 6 medicina-57-01113-t006:** The QPP questionnaire results between COVID-19 positive and negative pregnant women.

QPP (Mean ± SD)	COVID-19 (*n* = 168)	Non-COVID-19 (*n* = 136)	*p*-Value
Physical technical conditions	3.41 ± 0.53	3.72 ± 0.64	0.481
Medical technical competence	3.16 ± 0.50	3.94 ± 0.52	0.036
Identity oriented approach	3.64 ± 0.46	3.69 ± 0.54	0.822
Socio cultural atmosphere	3.08 ± 0.47	3.43 ± 0.50	0.741
Total satisfaction score	3.75 ± 0.62	3.82 ± 0.66	0.863

## Data Availability

Data available on request.

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
