# Peer review of "Coping Strategies and Health-Related Quality of Life in Pregnant Women with SARS-CoV-2 Infection"

_medicina, 2021, doi:10.3390/medicina57101113_

Round 1
Reviewer 1 Report
This is a well presented cross-sectional study among 304 pregnant women who completed standarized forms. Authors have revealed that unemployed pregnant women from a low economic status in rural areas undergo higher infection rates of SARS CoV-2 during pregnancy. Pregnant women with COVID-19 displayed statistically significant higher rates of anxiety and depression. Nonetheless, no differences were found among COVID-19 and non-COVID-19 units. Thus, authors conclude that the findings are not influenced by hospital care and efforts should be focused on a strong physical and psychological support on pregnant women, particularly with COVID-19.
In the materials and methods section, it would be appropriate to specify the setting of the study (Hospital and location).
Author Response
Dear reviewer,
We are thankful to receive your positive feedback, as much as we appreciate the time invested into helping us improve our manuscript for the benefit of science and future readers. Please see below the changes we made:
- We have described the study setting and location at lines 82-84.
- We have added in the Materials and Methods, at lines 93-99, a description of vaccination status among our patients, including the gynaecologists and government recommendations for pregnant women.
- We did some adjustments in the Discussion part, comparing a similar study held in the UK, with our results. This can be seen between lines 294 and 307.
- The references were formatted according to the journal’s standards.
Best regards,
The authors
Reviewer 2 Report
This study assesses the psychological impact of COVID-19 on pregnant women using standardised questionaires. The authors report that negative psychological impact was most evident with unemployed women living in poverty in rural areas. I have the following comments
- In the introduction section a separate paragraph should be added to describe the role of vaccination of pregnant women against COVID-19. What are the local recommendations? What is the percentage of vaccinated people in Romania? Do primary gynaecologists suggest vaccination to their patients?
- In the discussion section the authors mention a similar study conducted in the UK where no difference in anxiety levels was found between pregnant women with and without COVID. The authors should further explain possible differences between populations (UK and Romania). What is the reason for the a possible lack of trust in the health system?
Author Response

(The authors gave the same response as above.)
